# PIM Kinases in Multiple Myeloma

**DOI:** 10.3390/cancers13174304

**Published:** 2021-08-26

**Authors:** Jian Wu, Emily Chu, Yubin Kang

**Affiliations:** Division of Hematologic Malignancies and Cellular Therapy, Department of Medicine, Duke University Medical Center, Durham, NC 27710, USA; jw731@duke.edu (J.W.); emily.chu@duke.edu (E.C.)

**Keywords:** PIM kinase, multiple myeloma, resistance, inhibitor, PI3K/Akt/mTOR

## Abstract

**Simple Summary:**

Multiple myeloma is the second most common hematologic malignancy in the United States. Eventually, all myeloma patients will relapse and develop resistance to currently available agents. There is an unmet medical need to identify novel therapeutic targets. PIM kinases play an important role in myeloma pathogenesis and disease relapse. We herein provided a comprehensive review on the roles of PIM kinases in myeloma cell survival and proliferation and in the bone marrow microenvironment that supports myeloma growth. The development and testing of novel PIM kinase inhibitors were summarized. Finally, the preclinical studies of the combinatorial effects of PIM kinase inhibitors and other anti-myeloma agents were presented.

**Abstract:**

Multiple myeloma (MM) remains an incurable disease and novel therapeutic agents/approaches are urgently needed. The PIM (Proviral insertion in murine malignancies) serine/threonine kinases have 3 isoforms: PIM1, PIM2, and PIM3. PIM kinases are engaged with an expansive scope of biological activities including cell growth, apoptosis, drug resistance, and immune response. An assortment of molecules and pathways that are critical to myeloma tumorigenesis has been recognized as the downstream targets of PIM kinases. The inhibition of PIM kinases has become an emerging scientific interest for the treatment of multiple myeloma and several PIM kinase inhibitors, such as SGI-1776, AZD1208, and PIM447 (formerly LGH447), have been developed and are under different phases of clinical trials. Current research has been focused on the development of a new generation of potent PIM kinase inhibitors with appropriate pharmacological profiles reasonable for human malignancy treatment. Combination therapy of PIM kinase inhibitors with chemotherapeutic appears to create an additive cytotoxic impact in cancer cells. Notwithstanding, the mechanisms by which PIM kinases modulate the immune microenvironment and synergize with the immunomodulatory agents such as lenalidomide have not been deliberately depicted. This review provides a comprehensive overview of the PIM kinase pathways and the current research status of the development of PIM kinase inhibitors for the treatment of MM. Additionally, the combinatorial effects of the PIM kinase inhibitors with other targeted agents and the promising strategies to exploit PIM as a therapeutic target in malignancy are highlighted.

## 1. Introduction

Multiple myeloma (MM) is a hematologic malignancy characterized by the proliferation of malignant plasma cells. Precision medicine has heralded an era of change and challenge in the treatment of patients with MM. Although targeted therapies and immunotherapies have made significant advances in the personalized treatment of MM, clinicians still face the persistence of disease recurrence and drug resistance. The acquisition of anti-cancer drug resistance is a major issue with therapies in MM. Cancer cells utilize multiple intercellular and intracellular signaling cascades mediated by oncogenes such as PIM kinases to maintain cell growth and survival. In normal cells, the activity of these kinases is tightly controlled, whereas their sustained activation promotes apoptotic resistance and uncontrolled proliferation in cancer cells [1]. The complexity of the kinase molecular signaling network along with its crosstalk with alternative oncogenic signaling pathways provides ample opportunities for MM to develop productive adaptive mechanisms. PIM kinase activation has been shown to play a significant role in this bypass signaling mechanism. A better understanding of PIM kinase synergism, in addition to other signaling pathways, is important to the development of PIM inhibitors and to provide the rationale of combination therapy to improve the treatment efficacy for patients with MM.

## 2. Background—Expression and Regulation of PIM Kinases

The PIM kinases (PIM1, PIM2, PIM3) are a family of serine/threonine kinases and were named for their mode of discovery as proviral common integration site in Moloney murine leukemia virus (mMuLV)-induced lymphomas. PIM1 is located on chromosome 17, PIM2 on the X chromosome, and PIM3 on chromosome 15. They share high sequence homology at the amino acid level; PIM1 and PIM2 are 61% identical and PIM1 and PIM3 are 71% identical. PIM kinase genes comprise 6 exons and are transcribed into mRNA transcripts by alternative splicing. Two isoforms of PIM1 with sizes 34 and 44 kDa with comparable kinase activities are generated by translation of its mRNA from alternative initiation sites. The three isoforms of PIM2 with sizes 34, 37, and 40 kDa are similarly generated. PIM3 has one isoform. PIM1 is highly expressed in human fetal hematopoietic tissues such as the liver, spleen, and bone marrow [2,3]. PIM2 is mainly expressed in lymphoid and brain tissues. PIM3 is overexpressed in breast, kidney, and brain tissue [4]. 

In contrast to the majority of other kinases, there have been no other regulatory post-translational modifications reported for PIM kinases [5]. PIM kinases constitutively adopt an active conformation due to the presence of an acidic residue in the A-loop (Asp 200 in PIM1, Asp 196, and possibly Asp 198 in PIM2) that forms a salt bridge with basic residues of the catalytic loop (C-loop) and thus mimics the phosphorylation of the serine or threonine residue [6]. This would suggest that PIM kinases are regulated predominantly at the transcriptional and translational levels. 

The abundance of the constitutively active PIM kinases is tightly regulated through the Janus kinase/signal transducer (JAK/STAT) pathway activator and the NF-κB pathway. STAT3 and STAT5 are known to bind to the PIM1 promoter, upregulating its transcription [7]. PIM1 binds to and activates the suppressor of cytokine signaling (SOCS) through phosphorylation to prevent activation of the JAK/STAT pathway, forming a classical negative feedback loop [8]. In addition, PIM kinases are critical downstream of ABL (Abelson) and FLT3 (FMS-related tyrosine kinase 3) oncogenes and are required in driving tumorigenesis. Constitutively active FLT3 signaling up-regulates PIM1 expression and PIM is a key contributor to FLT3-induced proliferative and anti-apoptotic pathways [9]. BCR-ABL plays an important role in mediated cell transformation through induced PIM1 expression via active STAT5. 

It is noteworthy that PIM1 mRNA transcripts have a short half-life because of the presence of multiple copies of the destabilizing AUUU (A) sequence in the 3′ untranslated region. In addition, PIM mRNAs contain long GC sequence-rich near the 5′UTR and hence are a “weak” transcript that requires cap-dependent translation [10]. It is also important to note that the phosphorylation of S6K and 4EBP1, substrates of mTORC1 signaling, increases after PIM2 signaling and facilitates cap-dependent translation [11]. 

The stability of the transcribed PIM proteins is the key regulator of PIM activity. Its stability is largely controlled through ubiquitination and proteasomal degradation. Members of the heat shock protein family have opposite functions in stabilizing PIM activity. Binding to Hsp70 induces ubiquitylation and proteasomal degradation of PIM1, while Hsp90 protects PIM1 from proteasomal degradation [12]. On the other hand, ETK tyrosine kinase phosphorylates PIM1 at Y218, which is located in the activation loop, increasing its activity [13]. In addition, dephosphorylation of PIM kinases by the serine/threonine phosphatase, PP2A, promotes their ubiquitination and subsequent proteasomal degradation. Therefore, even though PIM kinases are constitutively active and do not depend on post-translational modifications for their activity, phosphorylation and dephosphorylation can still affect PIM’s stability.

## 3. PIM Deletion Studies

Initial studies involving PIM1, PIM2, and PIM3 triple knockout mice found that these mice are viable but have a decrease in body size, suggesting PIM kinases might act as sensitizers for growth factor signaling pathways [14]. 

The depletion of PIM1 by RNA interference in mice and cancer cells of the human prostate diminished cell proliferation, survival, and tumorigenicity [15]. PIM1 knockdown reduced BCL2 expression, and dynamic BH3 profiling analysis revealed that PIM1 prevents mitochondrial-mediated apoptosis in breast cancer cells [16]. Knockdown of PIM1 reduced renal carcinoma cell proliferation, colony formation, migration, invasion, and angiogenesis, suggesting that PIM expression may be a carcinogenic event [17]. Activation of PIM2 in colorectal cells led to increased glucose utilization and aerobic glycolysis, as well as energy production. Knockdown of PIM2 expression decreased energy production in colorectal tumor cells and increased their susceptibility to apoptosis [18]. Knockout of PIM3 expression by small interfering RNAs reduced human hepatoma cell proliferation [19]. PIM3 knockdown not only caused growth inhibition but also altered the phosphorylation of STAT3, mTOR, AMPK in human liposarcoma cells [20]. These observations link the expression of PIMs with clinical significance, underscoring their role as tumor biology modifiers, and rationalize further clinical development of PIM kinase inhibitors in clinical application. 

## 4. PIM Kinase and Cancers

PIM kinases are constitutively active serine/threonine kinases that are overexpressed in hematological malignancies [21]. Expression of both PIM1 and PIM2 are elevated in diffuse large B-cell lymphoma (DLBCL), while PIM2 alone is most highly expressed in B-cell chronic lymphocytic leukemia, acute myeloid leukemia (AML), and MM [22]. Increased PIM3 expression is typically observed in solid tumors [23]. Overexpression of PIM kinases is observed in MM and plays an important role in mediating survival and proliferation of MM cells, by inhibiting apoptosis and inducing cap-dependent translation, modulate immunes cells, respectively [24] (Figure 1).

### 4.1. PIM Kinases in Cancer Cell Cycle

PIM kinases exert their oncogenic effects through the phosphorylation of different proteins that are involved in the cell cycle and proliferation. Proliferation is enhanced by PIM1 mediated phosphorylation of the cell cycle inhibitor p21waf1, leading to cytoplasmic sequestration of p21waf1 and its inability to interact with cyclin E/CDK2 in the nucleus [25]. Furthermore, PIM kinases mediate phosphorylation and inactivation of fork-head transcription factors, FOXO1a and FOXO3a, which are involved in p27kip1 transcriptional repression [26]. All three PIM kinases can phosphorylate p27 kip at Thr157 and Thr198, which allows binding to 14-3-3 protein resulting in p27 nuclear export and proteasome-dependent degradation [27]. Due to its activity in cell cycle regulation, PIM kinase plays an important role in regulating the proliferation of tumor cells. 

PIM kinases also promote cell cycle progression via direct phosphorylation and activation of the CDC25A and CDC25C phosphatases, as well as inhibition of the CDC25A inhibitory kinase c-Tak1 [28]. Numerous studies have demonstrated that pharmacological inhibition of PIM induced cell cycle arrest in multiple tumor types, indicative of the overlapping activity of PIM kinases [29]. 

### 4.2. PIM Kinases in Cancer Cell Survival 

One of the main mechanisms by which PIM kinases exert their anti-apoptotic effects is via regulation of Bcl-2 family members [30,31]. The Bcl-2 family is comprised of both pro-apoptotic proteins, such as BAD and BAX, and anti-apoptotic protein, such as Bcl-2 and Bcl-XL. PIM phosphorylates BAD at Ser112, which disrupts its association with Bcl-2 and promotes binding to 14-3-3 and retention in the cytosol. Eventually, the dissociation of BAD and Bcl-2 promotes anti-apoptotic activity. PIM is also implicated in the regulation of apoptosis via the c-Jun-N-terminal kinase (JNK) signaling pathway [1]. PIM1 directly phosphorylates Ask1 at Ser83, which decreases its ability to phosphorylate and activate its substrates JNK and p38 [32]. Other anti-apoptotic activities of PIM kinases include phosphorylation of murine double minute 2 homolog (MDM2) at serine 166 and 186 to prevent proteasomal degradation of p53 in mantle cell lymphoma [33].

### 4.3. PIM Kinases in Cancer Cell Metabolism

All three PIM kinases phosphorylate the intracellular domain of Notch1 (N1ICD) at Ser2152 and thereby stimulate the nuclear localization and transcriptional activity of N1ICD. In breast cancer cells, PIM-mediated phosphorylation of N1ICD balances cell metabolism, while its inhibition enforces glycolytic metabolism via interfering with the mitochondrial function [34]. Several studies describe a correlation between cellular glucose metabolism and tumorigenesis: to sustain energetic demands due to increased cell proliferation, cancer cells need to readjust their cellular metabolism [35]. Currently available studies have suggested that PIM1 expression is correlated with the enhanced metastatic potential of the tumor and can be predictive of tumor outcome following chemotherapy and surgery [36]. Knockout of PIM1 could lessen glucose consumption and decrease key enzymes of the glycolytic pathway in hepatocellular carcinoma [37]. These findings indicate the importance of PIM kinases in regulating cancer cell metabolism and promoting tumor progression.

### 4.4. PIM Kinases and Immune Modulation

The mechanisms by which PIM kinases modulate the immune microenvironment and regulate immunes cells and the effects of PIM kinase inhibitors on immunity have not been systematically described. However, several studies have shown that PIM kinases can modulate the immune microenvironment and regulate immune cells [38,39]. 

PIM kinases positively regulate glycolysis in T cells and inhibition of PIM kinases leads to reduced glycolysis, increased T cell persistence, and enhanced tumor control. Moreover, PIM kinase inhibition in T cells led to higher FOXO1 activity, which translates to a T central memory phenotype (TCM, CD44+CD62L+) when compared with the control (vehicle-treated) T cells [40]. PIM1/3 inhibition prevented CD4+ T cell proliferation by inducing a G0/G1 cell cycle arrest without affecting cellular survival [41]. PIM1/PIM2 mRNAs are selectively up-or down-regulated in CD4+ cells, which subsequently affects the T cell differentiation into Th1 or Th2 cells by IL-2, IFN-α, and IL-4 [42]. T cell differentiation could be modulated by the upregulation of PIM1 and PIM2 expression. PIM2 induced by FOXP3 is essential for Treg cell expansion and, conversely, PIM2 also inhibits the suppressive function of Treg cells by phosphorylating FOXP3. These findings indicate the complex roles of PIM2 in the regulation of Treg cells [43]. Recently, studies have shown that PIM kinases promote survival and immune escape in primary mediastinal large-B cell lymphoma through modulation of JAK-STAT and NF-κB activity [44]. Given the potential role of PIM kinases in regulating tumor immunity, some cancer patients may benefit from combination strategies of PIM inhibition with checkpoint inhibitors. 

## 5. Targeting PIM Kinases in MM

PIM kinase has emerged as an exciting new target for the treatment of MM. PIM kinases play significant roles in MM progress by preventing apoptosis and by promoting the proliferation and survival of myeloma cells. 

### 5.1. PIM Kinases and Myeloma Bone Microenvironment

An important role of PIM2 in the bone marrow microenvironment has been recently reported [45]. High PIM2 expression levels in bone marrow stromal cells (BMSCs) and osteoblast precursors increase the expression of inhibitory factors of osteoblastogenesis, such as IL-3, IL-7, Activin A, TNFα, and transforming growth factor β, and block osteoblastogenesis and differentiation. Furthermore, PIM kinase inhibition was able to resume osteoblastogenesis in vitro and prevent bone destruction while suppressing MM tumor progression in vivo. Thus, the upregulation of PIM2 in both MM cells and BMSCs in bone lesions is suggested to have a pivotal role in tumor growth and bone loss in MM. PIM2 inhibition may become an important therapeutic strategy to target the MM cell-bone marrow microenvironment interaction [46].

Although little is known about the regulation of PIM1 in the process of osteoclastogenesis, there is evidence to support the finding that PIM1 could activate NF-κB and NFATc1 expression, and function as a modulator in the process of RANKL-induced osteoclastogenesis [47]. This data provides a new clue to further study the regulatory mechanism of PIM1 in the bone microenvironment.

### 5.2. PIM Kinases, and Myeloma Cell Homing and Migration

MM is thought to originate from long-lived plasma cells that develop in the germinal center of lymphoid tissues [48]. Some studies have demonstrated the presence of a small number of circulating plasma cells in over 70% of patients with MM [49] and its association with a poor prognosis [50]. The migration of cells through the blood to the bone marrow niches requires active navigation through the process of homing. The SDF-1/CXCR4 axis has been implicated in the expansion and homing of myeloma cells since the inhibition of CXCR4 reduces MM homing (Figure 1). The MM bone marrow microenvironment is a hypoxic niche. The hypoxic bone microenvironment conditions increase PIM activity with inhibition of the ubiquitin-mediated proteasomal degradation of PIM [51].

A clear association has been established between PIM1 and CXCR4 and this association confers a worse prognosis [52]. The association suggests that targeting aberrant PIM activity by small molecules would be rather promising by its effects on interfering not only with self-renewal but also with migration and homing of cancer cells. PIM1 has been shown in AML to regulate homing and migration of leukemic cells, possibly via phosphorylation-mediated modification on Serine339 of CXCR4 [53]. Conversely, inhibition of PIM by the small molecule SEL24-B489 blocked migration and homing by reducing CXCR4 surface expression [54]. Similarly, an association between PIM1 and CXCR4 could also be seen in chronic lymphocytic leukemia [55]. It is therefore probable that PIM kinases may at least partially contribute to the upregulation of CXCR4 in the hypoxic microenvironment of MM and, by this mechanism, also contribute to migration and homing of cells in MM.

### 5.3. PIM Kinases, Myeloma Cell Proliferation, and Cell Cycle Regulation

PIM2 is responsible for proliferation and cell cycle regulation in MM. PIM inhibition results in a significant decrease of mammalian target of rapamycin C1 (mTOR-C1) activity, which is critical for cell proliferation. Tuberous sclerosis protein 2 (TSC2), a negative regulator of mTOR-C1, is a PIM2 substrate and PIM2 phosphorylates TSC2 on Ser-1798 and relieves the suppression of TSC2 on mTOR-C1 [56]. In addition, 4EBP1and S6K, substrates of mTORC1 signaling, are also phosphorylated by PIM2, facilitating cap-dependent translation and proliferation. Evidence from the preclinical work of MM using PIM inhibitors demonstrated that inhibition of this process plays a key role in the anti-myeloma activity of PIM kinase inhibitors [11]. 

PIM inhibitor, LGB321, has been shown to decrease phosphorylated TSC2 and mTOR-C1 activity. The thiazolidine class PIM inhibitor strongly inhibited phosphorylation of 4EBP1 and lowered c-MYC expression in MM cell lines [45]. However, pharmacological inhibition with SGI-1776 results in no change in apoptosis or cell cycle regulation but affect protein translation with reduced phosphorylation of 4EBP1 and P70S6K [21]. 

### 5.4. PIM Kinases and Myeloma Cell Anti-Apoptotic Activity

The bone marrow microenvironment has a dominant role in the upregulation of PIM2 in MM. BMSCs and osteoclasts (OCs) confer MM cell survival through various factors. BMSCs and OCs increase PIM2 expression in MM cells via the IL-6/STAT3 and NF-κB pathway, respectively. PIM2 is dependent on NF-κB, and the anti-apoptotic effect of PIM2 could be completely inhibited by NF-κB inhibitors [57]. The PIM inhibitors thiazolidine and PI3K inhibitor LY294002, cooperatively enhance MM cell death [45]. On the other hand, reduced PIM2 expression with short interfering RNA decreased MM cell viability even when coculture with BMSCs or OCs, confirming the anti-apoptotic role of PIM2 in MM [45]. 

### 5.5. PIM Kinases and Myeloma Cell Resistance to Therapy

PIM kinases play pivotal roles in tumor progression and anti-cancer drug resistance. In hematologic malignancies, co-administrated standard treatment with PIM kinase inhibitors has proved useful in overcoming resistance in preclinical models. For instance, a combination of PIM inhibitors with JAK2 inhibitor in myeloproliferative neoplasms (MPN) [58] and a combination with cytarabine in AML overcame drug resistance [59].

Another key mechanism by which PIM kinases exert their resistance to anticancer therapies is their increased expression under hypoxia. It has been found that PIM kinases are expressed due to hypoxia in a HIF-1 independent manner by altering mitochondrial transmembrane potential and the activity of caspases-3 and -9 [60]. Introduction of siRNAs for PIM1 re-sensitizes cancer cells to chemotherapy drugs under hypoxia conditions. Furthermore, a recent study found that bortezomib treatment increases PIM half-life by prevention of PIM proteasomal degradation and therefore, the inclusion of a PIM kinase inhibitor in a bortezomib-based regimen could be effective in MM treatment [61]. 

## 6. PIM Kinase Inhibitors

Given the important role of PIM kinases in regulating malignant transformation, PIM kinases have become an important target of antitumor drug development. PIM kinase inhibitors are mainly classified as benzofurans, indoles, oxadiazoles, pyrazines, pyrimidines, pyrroles, quinolines. Newly developed pan-PIM kinase inhibitors are currently being tested in clinical trials, and these compounds have shown some success in MM and hematopoietic cancers. Table 1 lists current PIM kinase inhibitors. 

### 6.1. PIM447

PIM447 is a potent and selective pan-PIM kinase inhibitor, derived from the tool compound LGB321. PIM447 reduced the phosphorylation of Bad on Ser112 without affecting the levels of total BAD [75]. In addition, PIM447 reduced the levels of the Bad-regulated antiapoptotic protein Bcl-2 family members, such as Bcl-xl. PIM2 modulated mTOR-C1 activity and promoted myeloma cell proliferation through phosphorylation of TSC2. PIM447 strongly inhibited the phosphorylation of TSC2. In addition, PIM447 reduced the phosphorylation of downstream mTORC1 targets such as 4EBP1 at Thr37/46, P70S6 at Thr389, and S6BP at Ser 235/236. Furthermore, PIM447 reduced the levels and the stability of total c-Myc. 

In vitro work in MM cell lines revealed PIM447 is cytotoxic for multiple myeloma cells and has overcome the resistance conferred by BMSCs and OCs. PIM447 increased the percentage of cells in the G0-G1 phase and decreased the proliferative phases (S and G2-M) of the cell cycle. The effect of PIM447 was also investigated ex vivo in primary myeloma cells isolated from BM samples from 10 patients with multiple myeloma. PIM447 induced apoptosis in myeloma cells with low to moderate toxicity in lymphocytes. In vitro work has also demonstrated PIM447 could overcome the protective effect conferred by the BM microenvironment on multiple myeloma cells. Furthermore, the effect of combining PIM447 with several standard-of-care treatments showed a very strong synergism in the treatment of MM. 

Results of a first-in-human Phase I study of PIM447 in relapsed/refractory MM (NCT01456689), which mainly enrolled the Caucasian patients, have recently been reported [76]. Patients with relapsed and/or refractory MM were enrolled to determine the maximum-tolerated dose (MTD) or recommended dose (RD), safety, pharmacokinetics, and preliminary anti-myeloma activity of PIM447. PIM447 was administered in escalating oral doses of 70–700 mg once daily (QD) on 28-day continuous cycles. Seventy-nine patients with a median of four prior therapies were enrolled. Seventy-seven patients (97.5%) had an adverse event (AE) suspected as treatment-related, with treatment-related grade 3/4 AEs being mostly hematologic. An MTD of 500 mg QD and an RD of 300 mg QD was established. In this study population, a disease control rate of 72.2%, a clinical benefit rate of 25.3%, and an overall response rate of 8.9% were observed per modified International Myeloma Working Group criteria. Median progression-free survival at the RD was 10.9 months.

In addition, results of a Phase I study of PIM447 in relapsed/refractory MM (NCT02160951) which enrolled only Japanese patients, have recently been reported [77]. The study included 13 patients (7 patients at 250 mg once daily and 6 patients at 300mg once daily). The sole dose-limiting toxicity observed was grade 3 QTc prolongation in one patient from the 300 mg cohort. The most common suspected PIM447-related adverse events (AEs) included thrombocytopenia (76.9%), anemia (53.8%), and leukopenia (53.8%). All patients experienced at least one grade 3 or 4 AE, most frequently thrombocytopenia or leukopenia (61.5%). The overall response rate was 15.4%, disease control rate 69.2%, clinical benefit rate 23.1%, and two patients had a partial response (one in each dose group). Two patients treated with 250 mg QD had a progression-free survival >6 months. PIM447 250 mg or 300 mg QD was tolerated in Japanese patients with R/R MM. Further studies are required to evaluate clinical outcomes of PIM447 in combination with other drugs for the treatment of MM.

Results of new pharmacological combinations of PIM447 with other anti-myeloma agents have demonstrated a synergistic effect in MM in a preclinical study [78]. This study showed that PIM447 in combination with the standard treatment pomalidomide + dexamethasone exerts a potent antitumor effect and significantly improves survival with respect to the standard treatment. Mechanism of action studies performed in vitro and on mouse tumor samples suggest that the combination PIM-Pd inhibits protein translation processes through the convergent inhibition of c-Myc and mTORC1, which subsequently disrupts the function of eIF4E. The MM pro-survival factor IRF4 is also downregulated after PIM-Pd treatment. These results could be the basis for new clinical trials for this all-oral combination in the treatment of MM patients. 

### 6.2. GDC-0339 (Compound ***16***)

GDC-0339, a potent, orally bioavailable, and well-tolerated pan-Pim kinase inhibitor was found to be efficacious in RPMI8226 and MM1.S human multiple myeloma xenograft mouse models and is a candidate for early drug development [72]. GDC-0339 combined with PI3K inhibition provided good results in vitro by decreasing IC50 and likely also by enhancing tumor growth inhibition in vivo. Looking forward, PIM inhibition may also be useful in combination with other pathway targets such as USP7 [79] or applied to other disease indications such as T-cell acute lymphoblastic leukemia subset [80] and triple-negative breast cancer [81].

### 6.3. JP11646

JP11646, a new PIM2-selective non-ATP competitive inhibitor, reduces PIM2 expression at the protein and mRNA level in MM1.S and U266 cell lines in vitro and in vivo [71]. Additionally, JP11646 inhibits PIM2 kinase activity and the phosphorylation of its downstream targets such as 4EBP1, BAD, and Mcl2 [71]. Treatment with JP11646 resulted in a reduction in tumor burden and significantly improved the survival of xenograft myeloma mice [71]. 

### 6.4. Other PIM Kinase Inhibitors

INCB053914 is a novel, potent, and selective adenosine triphosphate-competitive pan-PIM kinase inhibitor. INCB053914 inhibited cell proliferation and phosphorylation of downstream substrates in cell lines from multiple hematologic malignancies [82]. Effects were confirmed in primary bone marrow blasts from patients with acute myeloid leukemia treated ex vivo and in blood samples from patients receiving INCB053914 in an ongoing phase 1 dose-escalation study [22]. In vivo, single-agent INCB053914 inhibited Bcl-2 associated death promoter protein phosphorylation and inhibited tumor growth in a dose-dependent manner in acute myeloid leukemia and multiple myeloma xenografts. Additive or synergistic inhibition of tumor growth was observed when INCB053914 was combined with selective PI3Kδ inhibition, selective JAK1 or JAK1/2 inhibition, or cytarabine [22].

SMI-4a has been reported to be a PIM2-specific kinase inhibitor that can block the growth of precursor T-cells in both leukemia and lymphoma. In addition, SMI-4a could inhibit the mTOR pathway to up-regulate the MAPK pathway, and ultimately reduce leukemic cell growth in vivo [83]. Similar to SMI-4a, SMI-16a reduces the capacities of colony formation of MM cells and their tumorigenic activity in vivo under acidic conditions, and restores the anti-MM effects of Doxorubicin [74]. Additionally, SMI-16a was also shown to increase the cytotoxic effects of carfilzomib by inhibiting the PIM2 accumulation [84]. 

### 6.5. Combination Strategies for PIM Inhibitors

Immunomodulatory drugs (IMiDs), including thalidomide, lenalidomide. and pomalidomide, are one of the mainstays in the treatment of MM. Lenalidomide is among the most used IMiDs in the treatment of MM. Pan-PIM kinase inhibitors SGI1776 and CX6258 exhibit significant anti-myeloma activity. Combining a pan-PIM kinase inhibitor with lenalidomide resulted in synergistic myeloma cell killing without additional hematologic or hepatic toxicities in an in vivo myeloma xenograft mouse model. In terms of mechanism, treatment with a pan-PIM kinase inhibitor promoted increased ubiquitination and subsequent degradation of IKZF1 and IKZF3, two transcription factors crucial for the survival of myeloma cells. Combining a pan-PIM kinase inhibitor with lenalidomide led to more effective degradation of IKZF1 and IKZF3 in multiple myeloma cells as well as xenografts of myeloma tumors. On the other hand, treatment with a pan-PIM kinase inhibitor resulted in increased expression of CRBN, providing IMiDs with more CRBN targets to bind to. These data elucidate the mechanism of pan-PIM kinase inhibitor mediated anti-myeloma effect and the rationale for the synergy observed with lenalidomide co-treatment and justify a clinical trial of the combination of pan-PIM kinase inhibitors and lenalidomide for the treatment of multiple myeloma [85]. 

## 7. Conclusions and Future Direction

Novel therapeutic approaches for the treatment of MM are urgently needed [86]. As scientific discovery continues to push the envelope in defining our understanding of PIM kinases, the current era of therapeutics gives us hope that PIM kinase inhibitors may become available for the treatment of MM patients in the not-so-distant future. 

This review highlights the growing number of studies demonstrating that aberrant PIM kinase activity plays an important role in cancer cell proliferation and survival, and mediates the drug resistance mechanism that allows cancer cells to evade cell death and develop a more aggressive phenotype. These findings suggest that PIM kinase inhibition may prove to be a beneficial strategy for treating MM patients including patients who are resistant and non-responsive to current treatment. Combination approaches will be necessary for PIM inhibition to provide a significant survival benefit. 

The expression level of PIM correlates with disease stage and prognosis in MM. Results from early phase clinical studies of PIM447 are encouraging and demonstrate impressive single-agent anti-myeloma activities in heavily pretreated patients with relapsed/refractory MM. Since bortezomib inhibition strongly increases the accumulation of catalytically active PIM2 [87], this finding provides a strong rationale for combining PIM inhibitors with current commonly used anti-myeloma agents or with other pathway inhibitors, including PI3K/AKT/mTOR and MAPK/ERK inhibitors to overcome drug resistance. PIM kinases have emerged as important effectors and mediators to mTOR activity in hematologic malignancies [88]. PIM inhibition may have an added advantage over mTOR inhibition given the myriad of other MM processes PIM are involved in.

Pan-PIM inhibitors demonstrated antitumor activity as monotherapy in patients with relapsed MM with a good tolerance profile [89]. Collectively, these studies demonstrate the great potential of targeting PIM kinases in the treatment of MM. Several aspects warrant further investigation, including a) the effects and the mechanism of different PIM kinases inhibitor on immune cell function and b) the optimization of the combination strategy of PIM kinases inhibitor with other standard treatment in the treatment of MM. As our understanding of PIM kinases evolves, more PIM kinases inhibitors will be identified for cancer treatment. 

## Figures and Tables

**Figure 1 cancers-13-04304-f001:**
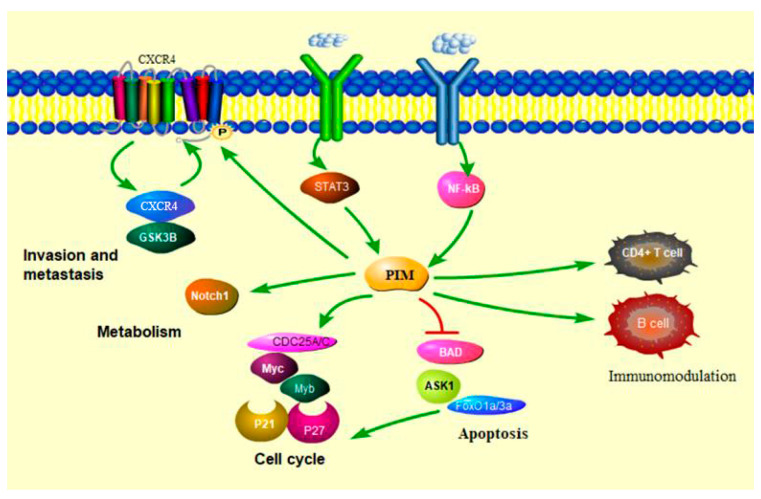
The regulation effect of PIM kinases in cancer. PIM kinase regulates many tumoral pathways by phosphorylating several target proteins, thereby activating or inactivating proteins involved in cell cycle progression, apoptosis, migration, or metabolism. PIM kinases promote the proliferation and survival of CD4+ T cells and B cells.

**Table 1 cancers-13-04304-t001:** PIM kinase inhibitors.

Name of Inhibitors	Class	IC50 or Ki	Preclinical or Clinical Studies
INCB053914	Adenosine triphosphate(ATP)-competitive inhibitor	PIM1: 0.24 nM PIM2: 30 nM PIM3: 0.12 nM	Preclinical [22]
LGB321	3-S-aminopiperidine pyridyl carboxamide	PIM1: 0.001 nM PIM2: 0.002 nM PIM3: 0.0008 nM	Preclinical [62]
LGH447 (PIM447)	3-S-aminopiperidine pyridyl carboxamide	PIM1: 5.8 pM PIM2: 18 pM PIM3: 9.3 pM	Phase I/II [63,64]
SGI-1776	Imidazopyridine	PIM1: 7 nM PIM2: 363 nM PIM3: 69 nM	Phase I [65]
IBL-202	Inhibitor of the PI3 kinases	NA	Preclinical [66,67]
SEL24-B489	FLT3-ITD inhibitor	PIM: 2 nM PIM2: 2 nM PIM3: 3 nM	Phase I/II [68]
AZD1897	Thiazolidine	PIM1: 3 nM PIM2: 3 nM PIM3: 3 nM	Preclinical [69]
AZD1208	Thiazolidine	PIM: 0.4 nM PIM2: 5 nM PIM3: 1.9 nM	Phase I (terminated) [70]
JP11646	Non-ATP competitive inhibitor	PIM1: 24 nM PIM2: 0.5 nM PIM3: 1 nM	Preclinical [71]
GDC-0339 (Compound **16**)	Diaminopyrazole	PIM1: 0.03 nM PIM2: 0.1 nM PIM3: 0.02 nM	Preclinical [72,73]
SMI-4a	benzylidene-thiazolidene-2,4-dione	PIM1: 21 nM PIM2: 100 nM	Preclinical [74]

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
