# Peer review of "PIM Kinases in Multiple Myeloma"

_cancers, 2021, doi:10.3390/cancers13174304_

Round 1
Reviewer 1 Report
Overall, this is a clear, concise, and well-written review on the role of PIM kinases in multiple myeloma. In this review, Wu et. al has done an excellent job of covering various aspects of PIM kinases including their role in cancer cells, their role in multiple myeloma, and their potential as therapeutic targets as single or combination agents.
There are a few grammatical corrections and a few comments that are advised to the authors. After these small corrections, the manuscript can be accepted for publication.

Reviewer 2 Report
In this review, Wu, Chu, and Kang provide an up-to-date view of PIM kinases in multiple myeloma, their involvement in various aspects of the myeloma cell biology such as cell cycle, survival, metabolism and immune effects, among others, and discuss the effects of different PIM kinase inhibitors as well as the current landscape of the same in the context of multiple myeloma. The structure and content of the review are generally appropriate, however there are some points that should be revised and corrected:
-In line 22 it should be indicated that LGH447 is the former name for PIM447.
-In line 41, do the authors refer to intercellular signaling cascades or, on the contrary, intracellular signaling cascades? The context of the sentence suggests that it refers to the second although the first is indicated.
-Line 119: the authors refer to PIM kinases as serine/threonine/tyrosine kinases. However, the majority of publications (including the authors themselves in the abstract) define PIM kinases as serine/threonine kinases. Please, clarify and unify the criteria.
-Lines 123-126: please, check this sentence. It is not clearly understood.
-Lines 143-145: Please, insert the reference referring to c-Tak1.
-Lines 203-204: osteoblast genesis should be substituted by osteoblastogenesis.
-In section 5.1, it is appropriate to include the next article: doi: 10.4049/jimmunol.1000885
-There seems to be an error in Figure 1. Please clarify if where you put CRCR4 it should put CXCR4
-In section 5.3, authors should mention the paper by Lu et al (PMID: 23818547) when they discuss about TSC2.
-Lines 243-245: it is needed to include the reference.
-Line 252: Is LY294002 a PIM kinase inhibitor or a PI3K inhibitor? Please, clarify and correct if necessary. Also, include the reference where the combination of thiazolidine and LY294002 is studied.
- Line 265: “under hypoxic” needs to be replaced by “under hypoxia”
-Lines 280-281: This phrase needs to be revised. According to the work by Paíno et al PIM447 reduces Bcl-xl but not Bcl-2 and Mcl-1
-Line 404: replace P13K for PI3K
